# Pharmacological evaluation of mangrove plant *Rhizophora mucronata* (Lam.) grown in the coastal area of Sundarbans

**Most. Chand Sultana Khatun[1,2], Md. Abdul Muhit[1], Md. Mahmudul Hasan Maruf[3], Md. Asif Iqbal[3], S. M. Abdur Rahman[1,4]***

1 Department of Clinical Pharmacy and Pharmacology, Faculty of Pharmacy, University of Dhaka, Dhaka, Bangladesh, 2 Department of Pharmacy, Mawlana Bhashani Science and Technology University, Tangail, Bangladesh, 3 Molecular Biology and Protein Science Laboratory, Department of Genetic Engineering and Biotechnology, University of Rajshahi, Rajshahi, Bangladesh, 4 Biomedical Research Center, University of Dhaka, Dhaka, Bangladesh

* smarahman@du.ac.bd

## Abstract

*Rhizophora mucronata*, a mangrove species native to coastal region of Bangladesh, has been drawn significant interests due to its potential ecological and therapeutic values, particularly as antioxidants, antibacterial and anti-inflammatory properties. The current investigation was aimed to identify the bioactive compounds from the different fractions of *Rhizophora mucronata* and to determine their antioxidants, analgesic, antidiabetic, antimicrobial along with *in-vitro* and *in-vivo* anticancer properties. Total three known compounds namely *N*-trans-para-caffeoyl-tyramine **(1)**, β-sitosterol **(2)** and rutin **(3)** were isolated from the ethyl acetate fractions (ERM) and their structures were elucidated by analyzing $^1$H-NMR spectral data. Dichloromethane (DRM) and ethyl acetate (ERM) fractions showed significant free radical scavenging properties ($IC_{50}$ value 12.18 and 11.7 µg/mL, respectively) compared to the standard ascorbic acid (6.36 µg/mL) in DPPH free radical scavenging assay. All three fractions exhibited notable analgesic effect in mice compared to standard drug diclofenac sodium in acetic acid induced writhing method. DRM and ERM fractions revealed significant glucose lowering effects compared to standard glibenclamide in streptozotocin induce diabetic mice model. Besides, all the fractions showed remarkable antibacterial effects (zone of inhibition 11.1–17.3 mm) against all selected *Gram-positive* but showed moderate activity against the *Gram-negative* bacteria. *In-vitro* cytotoxicity test of DRM and ERM fractions exhibited cytotoxic effect ($IC_{50}$ value 88.94 µg/ml and 127.6 µg/ml, respectively) against HeLa cell. The *in-vivo* cell growth inhibition of the three fractions on EAC (Ehrlich ascites carcinoma) cell demonstrated that ERM fraction furnished maximum cell growth inhibition (54.61%) compared to 84.83% inhibition by bleomycin. From the above findings, it is evident that ethyl acetate fractions of *R. mucronata*, can be exploited for future drug development and traditional medicinal applications.

**Data availability statement:** All relevant data are within the manuscript and its Supporting Information files.

**Funding:** The author(s) received no specific funding for this work.

**Competing interests:** No authors have competing interests.

## 1. Introduction

About 124 countries including Burma, India and South Africa contain mangrove ecosystem which consists of medicinally important plants [1]. Mangrove plants play a vital role in absorption of carbon to maintain productive and dynamic ecosystem. The ecosystem of coastal areas is also maintained by mangrove plants because of their capability to indulge the uttermost environmental factors such as high salinity, biofouling organism, high temperature, coastal food chain and inundation [2]. Apart from the ecological significance, mangrove plants have both nutritional and biological importance because of their rich source of proteins, fats, sugar, vitamins and minerals and the presence of bioactive metabolites [3]. Mangrove plants have enormous medicinal importance in Bangladesh [4]. In the ocean region of India and Bangladesh, around 55 species of mangroves from 22 genera are distributed which are known as different names such as swamps, tidal forests, tidal swamps or mangals [5].

*Rhizophora mucronata* (Lam.) (*R. mucronata*) is a small to medium size tree which is about 20–25 meters height usually found on the banks of river [6]. *Rhizophora mucronata* belongs to the family of Rhizophoraceae, largely distributed to tropical and subtropical coastal regions. Andaman and Nicobar are renowned for these mangrove plants that contain xanthone, lichixanthone along with atranorin, α-amyrin, β-amyrin, palmitone, β-sitosterol and dimyristyl ketone [7]. Diterpene rhizophorin B, C, D and E were also isolated from this mangrove [5,8]. Crude methanolic extract of *R. mucronata* has been reported to have anticancer properties in different cancer cell lines. [9]. 80% methanolic extract of *R. mucronata* leaves possesses flavonoids, gallic acid, quercetin and coumarin that has antiradical, antihyperglycemic and reduced lipid peroxidation activities. [10]. Detailed assessment of the traditional uses such as anticancer, anti-inflammation, antimicrobial, antidiabetic etc. along with 60 chemical components and their toxicity studies from *R. mucronata* has been reported earlier [11]. The antidiarrheal and anti-inflammatory activity guided isolation of bioactive components has led to isolate six compounds from *R. mucronata* bark extracts [12]. Tannins such as catechin, epigallocatechins were reported from the leaf extract of *R. mucronata* that relaxed or reduced the contractions of ileum in diarrheal rats. [13]. Old roots and leaves of *R. mucronata* are used for childbirth in Malayans, barks are used for the treatment of bloody urine. This plant is used by Chinese and Japanese people for diarrhea and Indochinese people uses it for angina [14].

Although enormous investigation on *R. mucronata* whole plant extracts has been accomplished previously, only few reports are found on different parts of the plant. Moreover, biological investigation on differential extracts or fractions depending on polarity of the solvents are missing. Besides, comprehensive pharmacological screening including *in vitro* and *in vivo* anticancer studies along with detailed understanding and characterization of bioactive components are very rare. Leaf extracts of *R. mucronata* were chosen because very little investigation in the pharmacological activity of leaf extracts have been reported. Moreover, studies indicated that the high concentration fruit flour of *R. mucronata* contains trace amounts of hydrogen cyanide (HCN) leading to create toxicity [15]. Alternatively, fresh leave extract of *R. mucronata* is considered safe and possessed no hazardous effect at standard therapeutic

concentration [16]. The current study has been designed to investigate various pharmacological activities such as antioxidants, analgesic and anti-inflammatory, antidiabetic, antimicrobial and cytotoxic potentials with different fractions (HRM, DRM and ERM) of *R. mucronata*. Moreover, in this research we focused on isolation and characterization of active constituents from the active fraction of *R. mucronata.*

## 2. Materials and methods

### 2.1. Materials and instruments

Ciprofloxacin, fluconazole, bleomycin, glibenclamide and ascorbic acid were obtained as generous gift from Eskayef Pharmaceutical Limited; Bangladesh and authorized suppliers supplied DME medium, 10% fetal bovine serum and HeLA cell was maintained by Center for Advance Research in Sciences (CARS) of University of Dhaka. To execute the cytotoxicity test, several instruments were used namely trinocular microscope with camera (Olympus, Japan), biological biosafety cabinet (NU-400E, Nuaire, USA), hemocytometer (Nexcelom, USA) and $CO_2$ incubator (Nuaire, USA). DMSO, 4' 6-diamidino-2-phenylindole (DAPI), trypan blue dye, sodium carbonate was purchased from Merk (Darmstadt, Germany).

For measuring $^1$H-NMR spectra (400 MHz), Burker AMX-400 was used. Coupling constants (*J*) are expressed in Hertz (Hz) and tetramethylsilane as internal reference was used to measure chemical shift expressed as in δ (ppm) scale. Pre-coated Silica plates (Silica gel 60 $F_{254}$- MerkKGa) were utilized to carry out thin layer chromatography (TLC) and plates were observed under UV light (254 nm) and finally sprayed with vaniline- sulfuric acid. Separation of compounds were performed by Column chromatography using Silica gel (Kieselgel 60, 230−400 mesh, Merck KGaA, Dermstadt, Germany) and size exclusion chromatography using Sephadex LH-20. Glass (20×20 cm) surface coated with silica gel slurry were used to prepare Preparative TLC plates.

### 2.2. Plant materials

Fresh leaves of the plant *Rhizophora mucronata* were collected from mangrove Sundarbans, Bangladesh in February 2019 upon taking verbal permission from the local forest officer at Karamjal, Sundarban area without disturbing its inhabitant properties The Principal Scientific Officer at Bangladesh National Herbarium, Dhaka has identified the exsiccated plant sample who provided a voucher specimen (accession number DACB-47468) for future reference.

### 2.3. Extraction and fractionization

Fresh leaves of *R. mucronata* were dried and grinded to make fine powder and 1 kg of powder was extracted with 100% methanol with occasional agitation. Filtration was accomplished initially by clean white cotton and then finally by the Whatman filter paper number 1. To obtain concentrated crude extracts, the solvent of the filtrate was evaporated by a rotary evaporator to yield 60 gm of *R. mucronata* (RM) methanol extract. The extract was fractionated by *n*-hexane, dichloromethane and ethyl acetate according to their increasing polarity [17], which furnished 4.5 g (0.45% yield) of *n*-hexane *R. mucronata* (HRM), 4.1 g (0.41% yield) of dichloromethane *R. mucronata* (DRM) and 7.2 g (0.72% yield) of ethyl acetate *R. mucronata* (ERM) extracts.

### 2.4. General phytochemical screening

Preliminary phytochemical analysis was performed to understand the types of constituents present according to the standard methods [18]. Presence of various plant secondary metabolites such as alkaloids, tannins, flavonoids, steroids, phenols, saponins, terpenoids, glycosides and quinones are screened using different standard chemical tests as described below.

**2.4.1. Test for alkaloid.** 2 M HCl was digested by 300 mg crude extracts (HRM, DRM, and ERM). After filtration, amyl alcohol was added to the acidic filtrate, a pink alcoholic layer was observed which indicate the presence of alkaloids.

**2.4.2. Test for saponin.** 5 ml water was added to the 300 mg extracts and boiled for 2 minutes. Mixers were cooled and mixed vigorously and left it for 3minutes. The presence of saponin was confirmed by formation of frothing.

**2.4.3. Test for flavonoids.** Small amount of the extracts was dissolved with methanol and added a few drops of concentrated hydrochloric acid and a very small amount of Mg ribbon to the solution. Immediate development of a red color indicated the presence of flavonoids.

**2.4.4. Test for Tannins.** Sodium Chloride was added to the aliquot of the extracts to create 2% strength. Then the mixers were filtrated and 1% gelatin was added to the filtrate. Precipitation stipulates the presence of tannins.

**2.4.5. Test for Triterpenes.** 5 ml chloroform was mixed to the 300 mg extracts and warmed for 30 minutes. Small volume of concentrated sulfuric acid was added to the chloroform mixed extract solutions and mixed properly. They gave red color which indicated the presence of triterpenes.

## 2.5. DPPH free radical scavenging assay

2,2-diphenyl-1-picrylhydrazyl (DPPH) free radical scavenging assay method described by Brand-Williams et al. [19] was used to examine the antioxidant activity of the crude extracts from *R. mucronata*. TLC plate was spotted with the diluted stock solution and appropriate solvent system was run over the plate. The compounds appeared at distinct position on the plate. Then 0.02% (w/v) DPPH in ethanol was sprayed over the plate at room temperature. The potential antioxidant property of the compounds was observed by changing the color due to reduction of DPPH by these compounds.

To assay potential antioxidant property of these fractions by scavenging DPPH free radical, the spectroscopic method was further applied [20]. At first serial dilution (500, 100, 50, 10, 5, 1 µg/mL) of the samples as well as ascorbic acid as standard was prepared with the volume 2 ml of each sample and 2 ml freshly prepared DPPH solution (0.004% w/v) were mixed carefully with each concentration of sample and ascorbic acid. Then all the solutions were incubated for 30 mins in a dark place at room temperature and UV absorbance of these mixers were recorded at 517 nm wavelength where methanol was used as blank solution. The percentage inhibition of DPPH scavenging was estimated by applying the following equation:

$$\text{Percentage Scavenging} = \frac{A_{blank} - A_{sample}}{A_{blank}} \times 100\%$$

Where $A_{blank}$ = absorbance of the methanol only containing DPPH and $A_{sample}$ = Absorbance of the test samples and ascorbic acid after reaction with DPPH solution. To get accurate result the experiment was accomplished for 3 times and percent of inhibition against sample concentration was plotted in a graph. $IC_{50}$ value (50% inhibition concentration) was computed from the graph.

## 2.6. Animals

Healthy adult locally bred Swiss albino mice, aged 4 weeks (body weight ranges from 30–40 gm) were collected from the North South University, Bangladesh. Adequate supply of food and water were ensured and kept them into a 12h darkness-light and ambient room temperature (25±2 $^0$C) for adopting the laboratory environment. Before oral delivery of any test samples the mice were kept into fasting conditions for at least 12-hour prior of experiment. All the animal related procedures were carried out following the institutional animal care handling protocol which was approved by refal committee of Faculty of Biological Sciences, University of Dhaka (Approval. number. 187/Biol. Sc.; December 20, 2022). Moreover, precautions were strictly taken to adhere with International Council for Laboratory Animal Science, the Nuffield Council on Bioethics (NCB), and the Council for International Organization of Medical Sciences (CIOMS/ICLAS) guidelines. All efforts were ensured to alleviate their sufferings. Finally. mice were euthanized by injecting phenobarbital sodium (125 mg/kg body weight) intraperitoneally. Death of the mice was confirmed by cessation of the chest movement and no response to toe pinch and finally the dead mice were disposed according to the institutional guidelines.

## 2.7. Analgesic test

Acetic acid induced writhing model of mice was used for analgesic test of the crude extracts HRM, DRM and ERM from *R. mucronata* [21]. A total of 25 mice were used for this experiment and divided into five groups (Group I to V); five animals encompass each group. Group I constituted control group which received 1% tween in saline as vehicle, Group II was given standard drug diclofenac-Na, Group III, IV and V received HRM, DRM and ERM fractions from *R. mucronata* respectively. The dose of standard drug diclofenac-sodium was 10 mg/kg i.p and that of the extracts were 200 mg/kg. 0.1 ml of 1% acetic acid was administered intraperitoneally for induction of peripheral pain after 30 minutes of administration of extracts. After 5 mins intervals the body contraction of mice referred to as writhing was counted for 15 minutes [22].

## 2.8. Anti-hyperglycemic test

Standard method was used to perform anti-hyperglycemic test of the three fractions HRM, DRM and ERM [23]. All the mice were divided into five groups (Group I to V) and five animals encompassed each group. Group I to V were treated with 0.5% methyl cellulose (control group), 5 mg/kg of glibenclamide (as reference drug), HRM, DRM and ERM fractions (dose of 200 mg/kg) respectively. After fasting for 16 hours, streptozotocin (60 mg/kg) dissolved in saline water was introduced intraperitoneally to all mice for induction of diabetes. Tail-vein blood of all group's mice was used for determination of blood glucose level by glucometer. Diabetes was noted if blood glucose level was higher than 11.5 mmol/L. All the mice were treated with the respective drug or extracts once daily for 7 days, and glucose levels of blood were determined by a glucometer at the 1st, 3rd, and 7th days.

## 2.9. Antimicrobial assay

The fractions HRM, DRM and ERM from *R. mucronata* were subjected to examine the antimicrobial activity using different strains by disc diffusion technique [24]. Antimicrobial assays were carried out in the laboratory of Biomedical Research Centre, University of Dhaka, Bangladesh. Five Gram-positive bacteria (*Bacillus cereus*, *Bacillus megaterium*, *Bacillus subtilis*, *Staphylococcus aureus*, *Sarcina lutea*) and eight Gram-negative strains (*Escherichia coli*, *Pseudomonas aeruginosa*, *Salmonella paratyphi*, *Salmonella typhi*, *Shigella boydii*, *Shigella dysenteriae*, *Vibrio mimicus*, *Vibrio parahemolyticus)* along with three unicellular fungi (*Candida albicans*, *Aspergillus niger*, *Saccharomyces cerevacae*) were used to perform the test. The fractions HRM, DRM and ERM were given at the dose of 500 µg/disc by a micropipette. For antimicrobial test, ciprofloxacin (5 µg/disc) and fluconazole (5 µg/disc) were used as the positive controls for antibacterial and antifungal activities respectively. The antimicrobial activity was determined by measuring the mean diameter of the zone of inhibition after conducting the experiment in thrice.

## 2.10. MTT cell viability assay against HeLa cell

The crude extracts HRM, DRM and ERM were subjected to perform *in-vitro* cytotoxicity test in human cervical carcinoma cell line (HeLa cell line) at Center for Advanced Research in Science of University of Dhaka [25]. Hela cells were maintained by DMEM (Dulbeco's modified eagles' medium) contained by 1% penicillin-streptomycin (1:1), 0.2% gentamycin, and 10% fetal bovine serum (FBS). Bleomycin was used as positive control with a series of concentration (62.5, 125 and 250- µL/mL) to estimate the cell proliferative test. 96-well plates with $2 \times 10^4$ cells per well (100 µL) were made to implant HeLa cells and a humidified atmosphere of 5% of $CO_2$ at $37^0C$ conditions were maintained to incubate the cells for 24 hrs. Hence, at first test samples (HRM, DRM and ERM) at the concentration (62.5, 125, 250 and 500 µL/mL) were dissolved in 2.5% DMSO, then incubation was done for 48 hr while 2.5% DMSO was used as negative control to compare the effect. A non-radioactive colorimetric cell proliferation and cytotoxicity assay kit (Sigma-Aldrich, USA) called Kit-8 was used for counting the cell viability [26]. This test was performed three times for each sample.

### 2.11. *In-vivo* anticancer test against Ehrlich Ascites Carcinoma (EAC) cells

**2.11.1. Transplantation of tumor.** Ehrlich Ascites Carcinoma (EAC) cells were acquired from Indian Institute of Chemical Biology, Kolkata, India. For culture and aspiration of EAC cells the standard published technique was followed [27]. To examine *in-vivo* cytotoxicity effect, cultured EAC cells were collected from 6–7 days aged Ehrlich ascites tumor-bearing mice. Then 0.9% normal saline was used to dilute the collected EAC cell and approximately $1 \times 10^6$ cells/ml was adjusted by a hemocytometer after dilution. About 0.1 ml of tumor cell suspension which contained $1 \times 10^6$ tumor cells was given intraperitoneally to each test animal.

**2.11.2. *In-vivo* cell growth inhibition.** In-vivo cell growth inhibition was carried out following the established protocol as described earlier [27]. Five groups of *Swiss albino* mice (n = 5) were used for this study. On the first day, all mice were injected with about $1 \times 10^6$ cells intraperitoneally. After 24 h of tumor inoculation, treatment was carried out for five days. Group I was negative control group which received only 10% DMSO, Group II was positive control which received bleomycin at a dose of 0.3 mg/kg/day; while Group III, IV and V received 50 mg/kg/day HRM, DRM and ERM extracts respectively. On the sixth day, all the mice were sacrificed, and aspiration of total tumor cell was done intraperitoneally by normal saline (0.98%). The viable EAC cells of both control and treated mice were identified by applying trypan blue and calculated with the help of hemocytometer. The following formula was used to calculate growth inhibition [28,29].

$$\text{Percentage Cell growth inhibition} = (1 - T/C) \times 100,$$

Where T = treatment group and C = control group.

### 2.12. Isolation of compounds from ERM fraction of *R. mucronata*

To isolate the pure compounds, 1 gm ERM fraction was subjected to column chromatography. A small amount of methanol solvent was used to dissolve the ERM fraction and silica gel was added and mixed properly. The dried sample was loaded on previously filled glass column with the slurry of silica gel (mesh size 60–120, 1g) and the column was run by a mixture of ethyl acetate: acetic acid: water (9:0.5:0.5) as the mobile phase and eluent was monitored by TLC. Identical TLC fractions were pooled. After that the pooled fractions were re-chromatographed by preparative thin layer chromatography (PTLC) using *n*-hexane: ethyl acetate (7:3) as the solvent system to obtain yellow crystalline powder of compound **1** (27 mg), white amorphous powder of compound **2** (7 mg) and white crystal of **3** (10 mg).

### 2.13. Measurement of spectral data of the isolated compounds

1H-NMR spectra of the isolated compounds were recorded by a Burker AMX-400 instrument operating at 400 MHz. Coupling constants (*J*) are expressed in hertz (Hz) and tetramethylsilane as internal reference was used to measure chemical shift expressed in δ (ppm) scale.

**N-trans-para-caffeoyl-tyramine (1)** White amorphous powder; 1H-NMR (400 MHz, $CDCl_3$): δ 7.42 (1H, d, *J* = 15.5 Hz, H-7′), 7.05 (2H, d, *J* = 8.0 Hz, H-2, 6), 7.0 (1H, d, *J* = 1.4 Hz, H-2′), 6.88 (1H, dd, *J* = 8.0, 1.4 Hz, H-6′), 6.81 (1H, d, *J* = 8.0 Hz, H-5′), 6.75 (2H, d, *J* = 8.0 Hz, H-3, 5), 6.30 (1H, d, *J* = 15.5 Hz, H-8′), 3.55 (1H, t, *J* = 7.0 Hz, H-7), 2.76 (1H, t, *J* = 7.0 Hz, H-8).

**β-Sitosterol (2):** White needle like crystals; 1H-NMR (400 MHz, $CDCl_3$): δ 0.83 (3H, d, *J* = 6.5 Hz, H-26), 0.87 (3H, m, *J* = 7.6 Hz, H-29), 0.92 (3H, d, *J* = 6.5 Hz, H-21), 0.95 (3H, d, *J* = 6.5 Hz, H-27), 0.98 (3H, d, *J* = 6.5 Hz, H-19), 1.1 (3H, s, H-18), 1.66–1.88 (m, H-7 and H-8), 3.32 (1H, m, H-3), 5.32 (1H, m, H-6).

**Rutin (3)** Yellow crystalline powder; 1H-NMR (400 MHz, $CDCl_3$): δ 6.26 (1H, d, *J* = 1.8 Hz, H-6), 6.45 (1H, d, *J* = 1.8 Hz, H-8), 6.68 (1H, d, *J* = 8.0 Hz, H-5′), 7.67 (1H, dd, *J* = 8.0 Hz, 1.5 Hz, H-6′), 7.71 (1H, d, *J* = 1.5 Hz, H-2′), 5.14 (1H, d, *J* = 8.0 Hz, H-1″), 3.61 (1H, m, H-2″), 3.70 (1H, dd, *J* = 12.0, 4.9 Hz, H-6″b), 3.88 (1H, dd, *J* = 11.0, 1.3 Hz, H-6″a), 4.56 (1H, d, *J* = 1.8 Hz, H-1‴), 3.64 (1H, m, H-2‴), 3.51 (1H, m, H-3‴), 3.67 (1H, m, H-4‴), 1.15 (3H, d, *J* = 6.7 Hz, H-6‴).

## 2.14. Ethical permission

Animal study was approved by the Ethical Review committee of Faculty of Biological Sciences (December 20, 2022, Ref. number. 187/Biol. Sc.), University of Dhaka.

## 2.15. Statistical analysis

All experiments were accomplished three times and represented as mean ±SEM. One-way ANOVA followed by Dunnet test using SPSS 16 software to carry significance test between control and treatment group where $*p$ value $< 0.05$ was considered as statistically significant.

## 3. Results

### 3.1. General phytochemical screening of the extracts

The fractions of HRM, DRM and ERM from *Rhizophora mucronata* were subjected for phytochemical analysis which demonstrated the presence of alkaloids, saponins, flavonoids, tannins and triterpenoids in all the extracts.

### 3.2. Antioxidant effect of crude extracts of *R. mucronata*

The results of antioxidant effect measured by DPPH free radical scavenging method are presented in Fig 1 and S1 Table. The percentage of DPPH scavenging activity was plotted against various concentrations of the samples and $IC_{50}$ for all the extracts and standard was measured. All the fractions exhibited promising antioxidant effect, among them DRM and ERM fractions showed high free radical scavenging properties ($IC_{50}$ value 12.18 and 11.70 μg/mL respectively) compared to the $IC_{50}$ value of standard drug ascorbic acid which was 6.36 μg/mL.

### 3.3. Analgesic activity of the fractions HRM, DRM and ERM of *R. mucronata*

The peripheral antinociceptive activity of the crude extracts HRM, DRM and ERM was accomplished by acetic acid induced writhing model in mice. The fractions HRM, DRM and ERM of *R. mucronata* revealed prominent analgesic activity at 200 mg/kg bw having writhing inhibitions 51.98%, 65.34% and 56.25% respectively compared to 81.53% inhibition showed by standard drug diclofenac-sodium (Group-II) at 10 mg/kg bw dose as shown in Table 1.

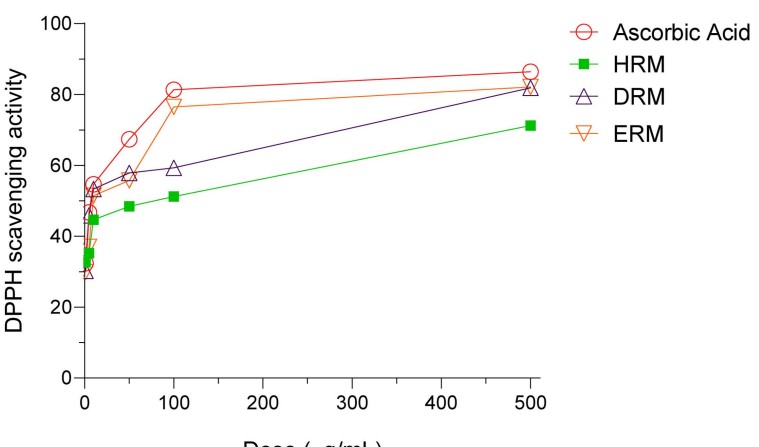 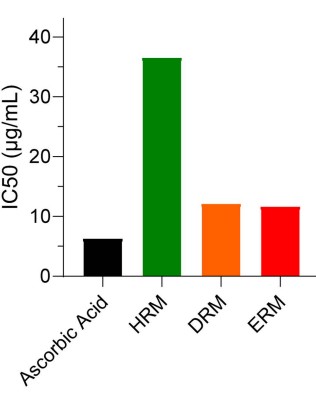

**Fig 1. Antioxidant activity of different fractions of *R. mucronate*.** Percentage inhibition (scavenging) of DPPH was plotted against concentrations of samples. Data were recorded for three times (S1 Table) and presented as mean SEM. $IC_{50}$ was determined by identifying the concentration at which 50% inhibition occurred.

**Table 1. Analgesic activity of HRM, DRM and ERM using acetic acid induced writhing model in mice.**

| Groups | Dose (mg/kg) | No of Writhing | % of Protection |
|---|---|---|---|
| Group-I | Vehicle | 35.2±2.7 | – |
| Group-II | 10 | 6.5±1.43* | 81.53 |
| Group-III | 200 | 16.9±5.2* | 51.98 |
| Group-IV | 200 | 12.2±4.33* | 65.34 |
| Group-V | 200 | 15.4±6.33* | 56.25 |

Values were expressed in Mean±SEM, (n=5). Group-I received 1% Tween 80 in saline and Group-II received 10 mg/kg Diclofenac-Na, Group-III, Group-IV and Group -V received HRM, DRM and ERM at the dose of 200 mg/kg body weight. *p<0.05 indicates significance compared with control group.

### 3.4. Antihyperglycemic test results of the *R. mucronata* fractions

The crude extracts HRM, DRM and ERM manifested prominent antihyperglycemic effect. All the fractions exhibited antihyperglycemic effects from day 1 to day 7 as observed from the line diagram (Fig 2). After 7 days, DRM and ERM fractions (Group IV and group V) exhibited significant glucose lowering effect of 60.73% and 52.17% respectively (at the dose level of 200 mg/kg body weight) which is comparable to the effect obtained by standard drug glibenclamide (Group II, 71.5% hypoglycemic effect). The results are statistically significant (Fig 2, S2 Table).

### 3.5. Antimicrobial activity of different fractions of *R. mucronata*

The antimicrobial efficacy of different fractions of HRM, DRM and ERM of *R. mucronata* is summarized in Table 2 and Fig 3. This investigation enumerated that all the three fractions displayed strong inhibitory effect (zone of inhibition 11.1–17.3 mm) against all selected gram-positive bacteria and showed moderate antibacterial activity against the gram-negative bacteria (zone of inhibition: 8.3–12.9 mm). However, HRM, DRM and ERM did not show any antifungal activity. The standard drug Ciprofloxacin showed reasonably higher antibacterial activity as shown in Fig 3 (Zone of Inhibition: 12.4–23.3 mm).

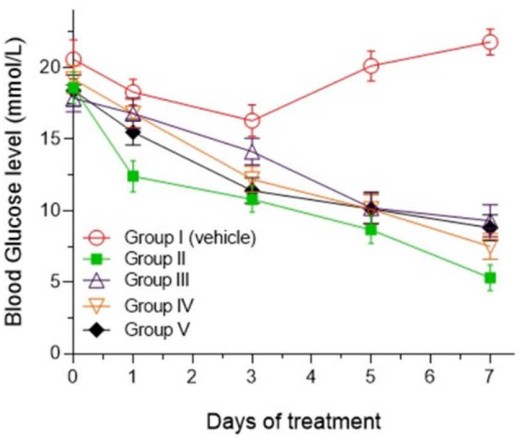
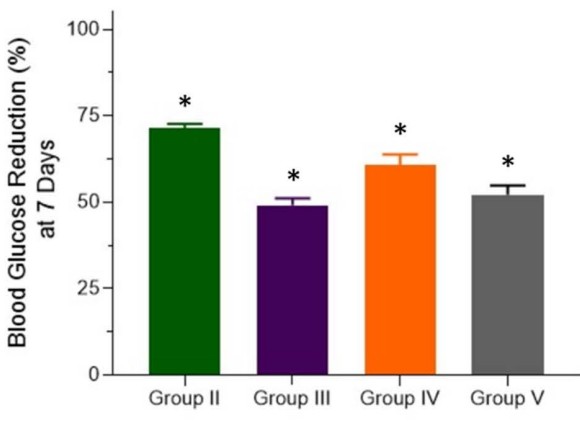

**Fig 2. Antidiabetic effect of fractions HRM, DRM and ERM on streptozotocin induced diabetic mice.** Values were expressed in Mean±SEM. 0.5% Methyl cellulose was received by the diabetic control group (Group I) and glibenclamide 5 mg/kg was received by reference group (Group II) and HRM, DRM and ERM were given to Group-III, Group-IV and Group-V respectively at the dose of 200 mg/kg. *p<0.05 indicates significance compared with control group.

**Table 2. Antimicrobial activity of test samples of *R. mucronata*.**

| Pathogens | Zone of Inhibition (mm) | | | |
|---|---|---|---|---|
| | HRM | DRM | ERM | Ciprofloxacin |
| **Gram positive bacteria** | | | | |
| *B. cereus* | 14.1±0.22 | 13.6±0.33 | 15.3±0.44 | 20.3±0.44 |
| *B. megaterium* | 14.3±0.12 | 15.1±0.11 | 13.3±0.21 | 22.3±0.21 |
| *B. subtilis* | 15.3±0.2 | 16.2±0.23 | 17.3±0.11 | 23.3±0.11 |
| *S. aureus* | 14.2±0.1 | 14.7±0.44 | 13.6±0.22 | 21.0±0.45 |
| *Sarcina lutea* | 12.1±0.22 | 11.1±0.22 | 12.7±0.11 | 23.0±0.22 |
| **Gram negative bacteria** | | | | |
| *E. coli* | 11.2±0.65 | 10.3±0.32 | 11.8±0.21 | 21.2±0.23 |
| *P. aeruginosa* | 10.4±0.77 | 8.7±0.22 | 9.3±0.22 | 12.4±0.14 |
| *S. paratyphi* | 9.3±0.11 | 10.3±0.41 | 8.3±0.51 | 18.5±0.44 |
| *S. typhi* | 8.7±0.66 | 12.9±0.55 | 11.4±0.42 | 17.7±0.14 |
| *S. boydii* | 11.3±0.26 | 8.7±0.32 | 8.6±0.48 | 18±0.18 |
| *S. dysenteriae* | 10.1±0.24 | 9.6±0.23 | 9.7±0.56 | 19.5±0.33 |
| *V. mimicus* | 9.9±0.67 | 10.2±0.11 | 10.2±0.33 | 18.3±0.42 |
| *V. parahemolyticus* | 10.2±0.12 | 9.6±0.33 | 10.5±0.45 | 20.0±0.41 |
| **Fungi** | | | | **Fluconazole** |
| *Candida albicans* | -- | -- | -- | 22.0±0.45 |
| *Aspergillus niger* | -- | -- | -- | 21.0±0.31 |
| *Sacharomyces cerevaceae* | -- | -- | -- | 23.0±0.33 |

Values were expressed in Mean±SEM (n=3). Dose of ciprofloxacin and fluconazole were 5 µg/disc whereas dose of fractions were 500 µg/disc.

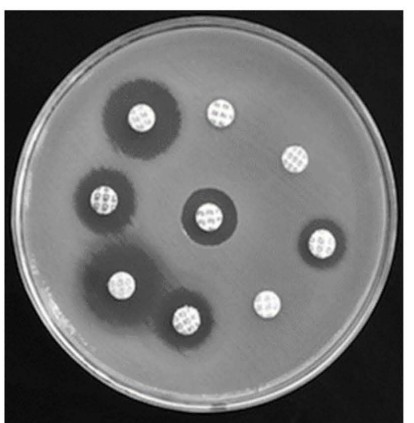 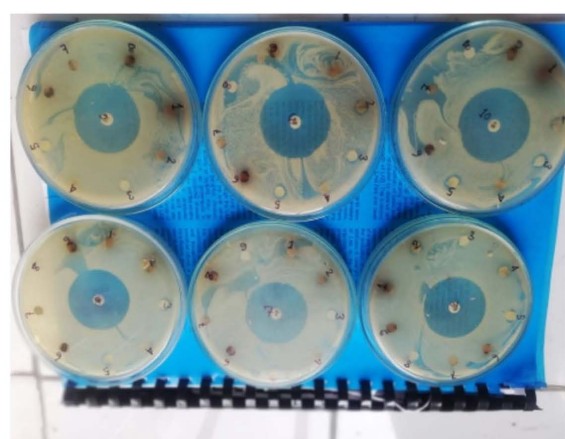

**Fig 3. Zone of inhibition of different fractions of *R. mucronata*.**

### 3.6. *In-vitro* cytotoxic activity of HRM, DRM ERM against HeLa cell

The investigation of *in-vitro* cytotoxic activity of different crude extracts against HeLa cell is illustrated in Fig 4. Among the different extracts DRM fractions showed the highest inhibitory effect with the $IC_{50}$ value of 88.94 µg/ml against HeLa cell. ERM fractions also exhibited good cytotoxic effect ($IC_{50}$ value of 127.6 µg/ml) against HeLa cell. Standard anticancer drug

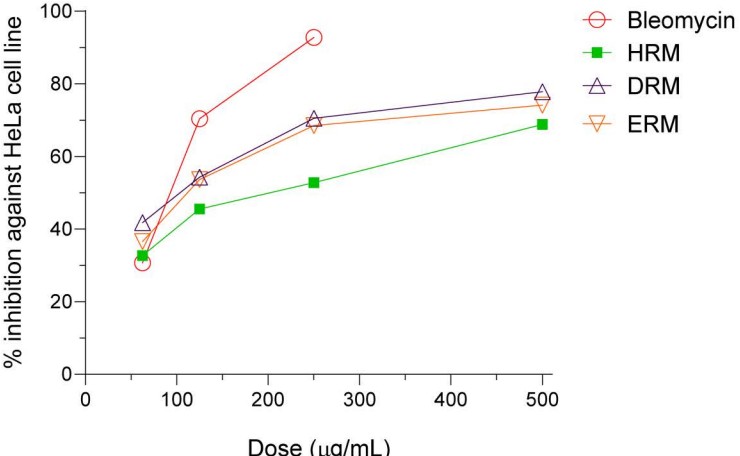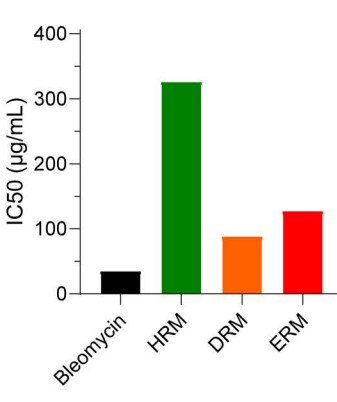

**Fig 4. *In -vitro* cytotoxic activity of bleomycin, HRM, DRM and ERM in Hella cell line.** The line diagram shows the percentage cell death and the bar diagram represents the $IC_{50}$ values of different extracts.

Bleomycin exhibited potential cytotoxic effect with an $IC_{50}$ value of 18.74 µg/ml. The morphological status of the cell is presented in S1 Fig.

### 3.7. *In-vivo* cell growth inhibition of the extracts HRM, DRM and ERM

*In-vivo* anticancer activity of the extracts HRM, DRM and ERM and standard drug bleomycin were examined in mice model and the results are shown in Table 3. Among the extracts, the maximum cell growth inhibition was given by the extracts ERM which was 54.61% and the rest of the extracts also displayed notable EAC cell growth inhibition (42.41% and 48.38% inhibition by HRM and DRM respectively) compared to standard drug bleomycin which showed 84.83% cell growth inhibition at 0.3 mg/kg body weight.

### 3.8. Identification and characterization of isolated compounds

As the ERM fraction was found to possess the highest anticancer activity *in vivo* test, initially it was selected for phytochemical screening. ERM fraction after column chromatographic technique furnished three pure compounds **1, 2** and **3** as presented in Fig 5. The structures of the isolated compounds were determined by studying 1D spectral feature and compared with the published data. The compounds were finally characterized as *N*-trans-para-caffeoyl-tyramine (**1**), β-sitosterol (**2**), and rutin (**3**).

**Table 3. *In-vivo* cell growth inhibition of the extracts HRM, DRM and ERM against EAC cell.**

| Group | Dose (mg/kg b.wt/day) | Viable EAC cell on day 5 after inoculation (x10⁶cells/ml) | % of cell growth inhibition |
|---|---|---|---|
| Group I | Control | 310±31.2 | – |
| Group II | Bleomycin (0.3 mg/kg) | 47±22.5* | 84.83 |
| Group III | HRM (50 mg/kg) | 178.5±30.2* | 42.41 |
| Group IV | DRM (50 mg/kg) | 160±28.3* | 48.38 |
| Group V | ERM (50 mg/kg) | 140.7±52.5 * | 54.61 |

Bleomycin, HRM, DRM and ERM significantly inhibit the growth of tumor cells at the doses of 0.3 and 50 mg/kg respectively.

Data are expressed in mean ±SEM at significant value of $p < 0.05$ and compared with control group. *$p < 0.05$ indicates significance compared with control group.

**Fig 5. Structure of isolated compounds (1-3) from *Rhizophora mucronate*.** Compound **1**: *N*-trans-para-caffeoyl-tyramine [30], compound **2**: β-sitosterol [31] and compound **3**: rutin [32]. The NMR Spectra of the compounds are available in the Supporting information.

Compound **1** was isolated from ERM fraction of *R. mucronata* by silica gel column chromatography as a white amorphous powder which was soluble in methanol. The TLC result furnished reddish black color when sprayed with vanillin-sulfuric acid followed by heating and the $R_f$ value was 0.37 in ethyl acetate: *n*-hexane (7:3) solvents system. The NMR spectrum of compound **1** is presented in the Supporting Information (S2 Fig). The resonances at δ 7.0 (1H, *d, J* = 1.4 Hz, H-2′), δ 6.88 (1H, dd*, J* = 8.0, 1.4 Hz, H-6′) and δ 6.81 (1H, d*, J* = 8.0 Hz, H-5′) in the [1]H NMR spectrum of compound **1** corresponding to the three ABX systems aromatic protons for caffeoyl moiety. The characteristic peak at δ 7.05 (2H, d, *J* = 8.0 Hz, H-2, 6) and δ 6.75 (2H, d, *J* = 8.0 Hz, H-3, 5) are corresponded to four protons of an AA′BB′ aromatic system. Two doublets resonance at δ 7.42 (1H, d, *J* = 15.5 Hz, H-7′) and δ 6.30 (1H, d, *J* = 15.5 Hz, H-8′) were assigned for C-7′ and C-8′ indicated that olefinic proton of C-7′ makes trans coupling with the olefinic proton of C-8′.Two coupled triplets for the methylene proton at δ 3.55 (1H, t, *J* = 7.0 Hz, H-7) and δ 2.76 (1H, t, *J* = 7.0 Hz, H-8) represented that compound **1** contains a tyramine moiety (S2 Fig). Based on the above spectral evidences compound **1** was confirmed as *N*-trans-para-caffeoyl-tyramine as compared with the published document [30].

Compound **2** was isolated from the ERM fraction of *R. mucronata* (RM) by sephadex LH-20, a size exclusion chromatography as a white needle crystal which showed deep purple color after spraying by vanillin-sulfuric acid on TLC (Fig 5). The $R_f$ value was 0.65 in *n*-hexane: ethyl acetate (7:3) solvents system and the compound **2** was found to be soluble in chloroform. The multiplets at δ 3.32 ppm was assigned for H-3 in [1]H NMR spectra of compound **2** exhibited the posture and abundance which indicated steroidal center. The observation showed that the characteristic signal of olefinic H-6 of the steroidal skeleton was given at δ 5.32 (*t*, 1H) (S3 Fig). The spectrum also described that signal at δ 1.10 (*s*,3H) and δ

0.98 (*d*,3H) ppm corresponding to a double tertiary methyl group at C-18 and C-19, respectively. Based on spectral data, the identity of the compound **2** was confirmed as β-Sitosterol as compared with published report [31].

Compound **3** was obtained from ERM fraction of *R. mucronata* using sephadex LH-20, a size exclusion chromatography as a yellow crystalline powder. The TLC result gave yellow color when sprayed with vanillin-sulfuric acid followed by heating and the $R_f$ value 0.51 was in ethyl acetate: *n*-hexane (8:2) solvents system and the compound is soluble in methanol. The $^1$H NMR spectrum of the compound **3** exhibited a set of ABX-type aromatic protons [δ 7.71 (1H, d, *J* = 1.5 Hz, H-2'), 6.68 (1H, d, *J* = 8.0 Hz, H-5') and 7.67 (1H, dd, *J* = 8.0 Hz, 1.5 Hz, H-6')] along with a meta-coupled pair at δ 6.26 (1H, d, *J* = 1.8 Hz, H-6), 6.45 (1H, d, *J* = 1.8 Hz, H-8), representing a flavonol skeleton in its structure. The compound also showed two anomeric protons at δ 5.14 (1H, d, *J* = 8.0 Hz, H-1'') and 4.56 (1H, d, *J* = 1.8 Hz, H-1''') indicating the presence of a glucose and rhamnose unit with *O*-glycosidic linkages. Furthermore, two protons at δ 3.70 (1H, dd, *J* = 12.0, 4.9 Hz, H-6''b), 3.88 (1H, dd, *J* = 11.0, 1.3 Hz, H-6''a) reveals that rhamnose molecule is attached with C-6'' and another doublet at δ 1.15 (3H, d, *J* = 6.7 Hz, H-6''') , a characteristic signal of rhamnose methyl group at C-6''' is also observed (S4 Fig). Based on these spectral features and comparing with the published data, compound **3** was characterized as rutin, a flavonoid glycoside as shown in Fig 5 [32].

## 4. Discussion

The current research demonstrates the phytochemical and pharmacological evaluation of *R. mucronata* grown in Sundarbans. The different fractions such as HRM, DRM and ERM from *R. mucronata* exhibited several pharmacological properties such as antioxidant, analgesic, antidiabetic, antimicrobial and cytotoxic potentials. As shown in Fig 1, different extracts of *R. mucronata* exhibited prominent free radical scavenging activity. The ERM fraction showed the most prominent DPPH inhibitory effect ($IC_{50}$ value 10.41 μg/mL) compared to standard drug ascorbic acid ($IC_{50}$ value 6.36 μg/mL). A previous study reported that ethanol and methanol extracts of this species provided the most prominent free radical scavenging activity [33]. Therefore, the highest activity of the comparatively polar ERM fraction indicates that polar compounds might be responsible for antioxidant effect. Plants are rich sources of flavonoids and phenolic compounds which have the ability to scavenge free radicals due to their ability to donate hydrogen atom [34]. Phytochemical investigation of ERM fraction provided phenolic compound *N*-trans-para-caffeoyl-tyramine (**1**) and flavonoids rutin (**3**) which might have contributed for the strong antioxidant property of ERM fraction.

Peptic ulcer to gastric mucosal damage and perforation are the common symptoms due to non-steroidal anti-inflammatory drugs. Therefore, new drugs with very high safety margin are warranted. In this study the fractions HRM, DRM and ERM provided promising analgesic effects assessed by their excellent writhing inhibition (51.98 to 65.34% reduction in writhing) in mice model. In an earlier investigation, ethanolic extracts of *R. mucronata* showed 44% reduction in the writhing with a dose of 200 mg/kg body weight in mice which is in accordance with our findings. [35]. The current extracts from Sundarbans showed better writhing inhibition. Visceral pain sensation in abdomen is evoked by intraperitoneally injected acetic acid to produce the localized inflammatory response which enhance the pain mediators such as prostaglandin specially PGE2, PGF2α and therefore results writhing or abdominal constriction in mice [36]. Based on the results in all the fractions, we may assume that the analgesic action of the extracts might be due to synergistic effects of several components present in the extracts.

It has been also shown that the fractions HRM, DRM and ERM significantly decrease blood glucose level in streptozotocin induced type 2 diabetes mice. Our findings reestablish the previous findings where ethanolic fractions of *R. mucronata* significantly inhibited α-amylase and α-glucosidase enzymes activity as well as controlled glycemic index and reduced diabetic complications [37]. The blood glucose lowering effect of the three extracts may be due to rejuvenation of the β-cell and enhancing glucose tolerance of the cells [37]. It might be assumed that different fractions of *R. mucronata* contain several bioactive constituents which might stimulate beta cells to release insulin or may increase insulin sensitivity [38].

In 2022, about 5 million global deaths occur due to drug resistance infection and this will be about 10 million per year within 2050 because multidrug resistance to microbial strains reduces susceptibility to antibiotic [39]. Synthetic drugs and orthodox are avoided because of their unwanted side effects and therefore, herbal remedies are the choice to treat patient [40]. Recently, a great deal of attention is provided to the development of antimicrobials from natural sources [41]. Hence, we observed that the fractions HRM, DRM and ERM exhibited very impressive antibacterial activity against both gram- positive and gram-negative bacteria which is agreeable with an earlier report where leave extracts from *R. mucronata* showed potential antibacterial activity against *Bacillus subtilis*, *Staphylococcus aureus*, *Candida albicans*, *Aspergillus fumigatus* and *Aspergillus niger* and moderate activity against *Pseudomonas aeruginosa and Proteus vulgaris* [42]. The potent antimicrobial activity of the crude extracts may be due to the presence of phenols, flavonoids, alkaloids, steroids [43].

Cancer is the 2nd life threatening disease which has emerged as prime cause of death of many patients. According to WHO, about 9.6 million deaths occurred by cancer in 2018. The leading etiological factors that cause cancer are diets (chemical carcinogen), certain viral infection (biological carcinogen) and environment (physical carcinogen) [44]. The *in-vitro* cytotoxicity of crude extracts HRM, DRM, and ERM from *R. mucronata* revealed good inhibitory effect against HeLa cell with respectable $IC_{50}$ value of 235.04, 88.94 and 127.6 µg/ml respectively. Based on this *in-vitro* result, *in-vivo* anticancer activity of the extracts was accomplished against EAC cell induced carcinogenicity. In *in-vivo* test, ERM gave significant inhibitory effect which was 54.61% and the rest of the extracts also had good EAC cell growth inhibition compared to standard drug bleomycin which showed 84.83% cell growth inhibition. These results reinforce the findings of a previous *in vitro* anticancer study of *R. mucronata* crude methanol extracts where the extracts were found to be effective against colon, lung, prostrate and breast cancers [9]. The activity of the extracts might be due to interference in several biochemical pathways for DNA synthesis and cell growth or inactivation of NFkB which is responsible for cell differentiation and proliferation [45,–46]. In order to identify the chemical components responsible for potent cytotoxic effect of ERM, phytochemical screening was accomplished. Three compounds *N*-trans-para-caffeoyl-tyramine (**1**), β-sitosterol (**2**) and rutin (**3**) were isolated from ERM fraction of *R. mucronata* among which compound **1** and **3** were isolated from this plant for the first time. These three compounds might be responsible for exerting cytotoxic effects.

## 5. Conclusion

Methanolic extracts of *R. mucronata* have been fractionated to obtain three fractions based on varying polarity. In this study we have observed that the fractions HRM, DRM and ERM exerted prominent antioxidant, peripheral anti-nociceptive, antidiabetic and antimicrobial effects on selective pathogens. The outcomes of this investigation claim that the plant species can be used as a folk medicine. Here three compounds *N*-trans-para-caffeoyl-tyramine (**1**) and β-sitosterol (**2**) and rutin (**3**) were isolated from ERM fraction of *R. mucronata* and *N*-trans-para-caffeoyl-tyramine (**1**) and rutin (**3**) were isolated from this plant for the first time. The fractions also gave significant *in-vitro* cytotoxic effect on HeLa cell and *in-vivo* anticancer effect against EAC cell which were performed for the first time by this plant extracts. Further studies are necessary to identify the active constituents from different fractions, examine dose, pharmacodynamic, pharmacokinetic and other properties of the extracts as well as the active constituents.

## Supporting information

**S1 Fig. *In-vitro* cytotoxicity activity of control, standard and fractions at the concentration of 500 (µg/ml) observed under an inverted light microscope.**
(PDF)

**S2 Fig. 1H-NMR spectrum of compound 1 (N-trans-para-caffeoyl-tyramine).**
(PDF)

**S3 Fig. 1H-NMR spectrum of compound 2 (β-Sitosterol).**
(PDF)

**S4 Fig. 1H-NMR spectrum of compound 3 (Rutin).**
(PDF)

**S1 Table. DPPH scavenging activity of RM fractions HRM, DRM and ERM and ascorbic acid.**
(PDF)

**S2 Table. Antidiabetic effect of fractions HRM, DRM and ERM on streptozotocin induced diabetic mice.**
(PDF)

**S3 Table. *In-vitro* cytotoxic activity of HRM, DRM and ERM from RM on HeLa cell line.**
(PDF)

## Acknowledgments

The authors would like to express their sincere gratitude and thanks to the authorities of Molecular Biology and Protein Science Laboratory, University of Rajshahi. Besides, Most. Chand Sultana Khatun is gratefully acknowledged and thanks to the Ministry of Science and Technology for providing a fellowship to conduct this research.

## Author contributions

**Conceptualization:** Most. Chand Sultana Khatun, S. M. Abdur Rahman.

**Data curation:** Most. Chand Sultana Khatun, Md. Mahmudul Hasan Maruf, Md. Asif Iqbal.

**Formal analysis:** Most. Chand Sultana Khatun, Md. Abdul Muhit.

**Investigation:** Most. Chand Sultana Khatun, Md. Mahmudul Hasan Maruf, Md. Asif Iqbal.

**Methodology:** Md. Abdul Muhit, S. M. Abdur Rahman.

**Project administration:** Md. Abdul Muhit, S. M. Abdur Rahman.

**Resources:** S. M. Abdur Rahman.

**Supervision:** Md. Abdul Muhit, S. M. Abdur Rahman.

**Writing – original draft:** Most. Chand Sultana Khatun.

**Writing – review & editing:** Most. Chand Sultana Khatun, Md. Abdul Muhit, S. M. Abdur Rahman.

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
