## [Decision Letter · Decision Letter 0]

21 Oct 2025

Dear Dr. Rahman,

Thank you for submitting your manuscript to PLOS ONE. After careful consideration, we feel that it has merit but does not fully meet PLOS ONE’s publication criteria as it currently stands. Therefore, we invite you to submit a revised version of the manuscript that addresses the points raised during the review process.

We look forward to receiving your revised manuscript.

Kind regards,

Nadeem Nazurally, Ph.D

Academic Editor

PLOS ONE

Journal Requirements:

3. To comply with PLOS ONE submissions requirements, in your Methods section, please provide additional information regarding the experiments involving animals and ensure you have included details on (1) methods of sacrifice, and (2) efforts to alleviate suffering.

4. Please ensure that you include a title page within your main document. We do appreciate that you have a title page document uploaded as a separate file, however, as per our author guidelines (http://journals.plos.org/plosone/s/submission-guidelines#loc-title-page) we do require this to be part of the manuscript file itself and not uploaded separately.

Reviewers' comments:

Reviewer's Responses to Questions

**Comments to the Author**

1. Is the manuscript technically sound, and do the data support the conclusions?

Reviewer #1: Yes

2. Has the statistical analysis been performed appropriately and rigorously?

Reviewer #1: N/A

3. Have the authors made all data underlying the findings in their manuscript fully available?

Reviewer #1: Yes

4. Is the manuscript presented in an intelligible fashion and written in standard English?

Reviewer #1: Yes

Reviewer #1: 1) correct comma usage in English language - in abstract

2) item 2.2 Plant materials - also correct comma usage in English language

3) item 2.4 General phytochemical screening: Preliminary phytochemical analysis was performed to understand the types of constituents present according to the standard methods [could you describe more about it?].

4) item 2.5: "The % inhibition or percentage of DPPH scavenging was estimated by applying the following equation" [I suggest use of "percentage" - not both symbol and word]

5) Figure: for me the figures look blurry, needing more resolution

**Do you want your identity to be public for this peer review?** For information about this choice, including consent withdrawal, please see our Privacy Policy

Reviewer #1: **Yes:**  Fernando de Figueiredo Porto Neto

---

## [Author Response · Author response to Decision Letter 1]

14 Dec 2025

13.12.2025

Dear Editor:

Thanks to respected editor and the reviewers for your nice and constructive comments. According to the editorial and reviewer’s suggestions, we have revised the manuscript and are submitting a modified version of our manuscript. The revision is marked with a yellow color in the manuscript.

Editorial Response:

Major comments:

Author’s reply: Thank you for the generous suggestions. The manuscript has been revised accordingly. We have added the authors name, title and affiliations in the front page.

Author’s reply: Thank you. Usually collecting the common species from the nearby areas of sundarban’s does not require any official permission. However, we had taken a verbal permission from the local forest office which has been mentioned in the manuscript as follows:

Fresh leaves of the plant Rhizophora mucronata weredss collected from mangrove Sundarbans Bangladesh in February 2019 upon taking verbal permission from the local forest officer at Karamjal, Sundarban area without disturbing its inhabitant properties. (section 2.2)

3. To comply with PLOS ONE submissions requirements, in your Methods section, please provide additional information regarding the experiments involving animals and ensure you have included details on (1) methods of sacrifice, and (2) efforts to alleviate suffering.

Author’s reply: We have revised the manuscript and added a section (section 2.6) for handling animals. According to your suggestion, we have further included detail animal handling and sacrifice methods. The following section (section 2.6) has been included in the manuscript:

2.6 Animals

Healthy adult locally bred Swiss albino mice, aged 4 weeks (body weight ranges from 30-40 gm) were collected from the North South University, Bangladesh. Adequate supply of food and water were ensured and kept them into a 12h darkness-light and ambient room temperature (25±2 0C) for adopting the laboratory environment. Before oral delivery of any test samples the mice were kept into fasting conditions for at least 12-hour prior of experiment. All the animal related procedures were carried out following the institutional animal care handling protocol which was approved by ethical committee of Faculty of Biological Sciences, University of Dhaka (Approval. number. 187/Biol. Sc.; December 20, 2022). Moreover, precautions were strictly taken to adhere with International Council for Laboratory Animal Science, the Nuffield Council on Bioethics (NCB), and the Council for International Organization of Medical Sciences (CIOMS/ICLAS) guidelines. All efforts were ensured to alleviate their sufferings. Finally. mice were euthanized by injecting phenobarbital sodium (125 mg/kg body weight) intraperitoneally. Death of the mice was confirmed by cessation of the chest movement and no response to toe pinch and finally the dead mice were disposed according to the institutional guidelines.

4. Please ensure that you include a title page within your main document. We do appreciate that you have a title page document uploaded as a separate file, however, as per our author guidelines (http://journals.plos.org/plosone/s/submission-guidelines#loc-title-page) we do require this to be part of the manuscript file itself and not uploaded separately.

Author’s reply: Thank you. We have revised the manuscript accordingly.

5. Please include captions for your Supporting Information files at the end of your manuscript, and update any in-text citations to match accordingly. Please see our Supporting Information guidelines for more information: http://journals.plos.org/plosone/s/supporting-information

Author’s reply: Captions for supporting information have been added at the end of the manuscript

Author’s reply: This suggestion has been followed and no unnecessary reference has been included.

Reviewer 1 Responses:

1. Correct comma usage in English language - in abstract.

Author’s reply: Thank you for the suggestion. The abstract has been revised accordingly.

2. Item 2.2 Plant materials - also correct comma usage in English language.

Author’s reply: The comma usage in English language is corrected accordingly.

3. Item 2.4 General phytochemical screening: Preliminary phytochemical analysis was performed to understand the types of constituents present according to the standard methods [could you describe more about it?].

Author’s reply: Thank you for the suggestion. The following sentences has been mentioned in the manuscript:

Presence of various plant secondary metabolites such as alkaloids, tannins, flavonoids, steroids, phenols, saponins, terpenoids, glycosides and quinones are screened using different standard chemical tests. The description of all the chemical tests was also included in section 2.4.

4. Item 2.5: "The % inhibition or percentage of DPPH scavenging was estimated by applying the following equation" [I suggest use of "percentage" - not both symbol and word].

Author’s reply: Thank you. This has been revised accordingly (Section 2.5 and 2.11).

5. Figure: for me the figures look blurry, needing more resolution.

Author’s reply: Thank you. We have improved the resolution of the figures.

Reviewer 2 Responses:

In this paper, the authors report on the investigation of the anti-diabetic, antibacterial, anti-cancer and anti-oxidant properties of hexane, dichloromethane and ethylacetate extracts of the leaves of the mangrove R. Mucronata. In depth in vitro and in vivo studies were conducted. Three compounds were isolated from the extracts and successfully characterised by NMR.

Overall, the research methodology is sound. However, the authors failed to highlight previous studies related to the current one and they did not clearly point out the novelty of the current investigation. Therefore, the manuscript requires major revision before it can be considered for publication.

1. The abstract should be reformulated and better presented to readers to have a better flow and understanding of the study rationale and what investigations have been carried out highlighting key results only.

Author’s reply: Thank you for the insightful suggestion. Abstract has been revised accordingly as shown below:.

Rhizophora mucronata, a mangrove species native to coastal region of Bangladesh, has been drawn significant interests due to its potential ecological and therapeutic values, particularly as antioxidants, antibacterial and anti-inflammatory properties. The current investigation was aimed to identify the bioactive compounds from the different fractions of Rhizophora mucronata and to determine their antioxidants, analgesic, hypoglycemic, antimicrobial along with in-vitro and in-vivo anticancer properties. Total three known compounds namely N-trans-para-caffeoyl-tyramine (1), β-sitosterol (2) and rutin (3) were isolated from the ethyl acetate fractions (ERM) and their structures were elucidated by analyzing 1H-NMR spectral data. Dichloromethane (DRM) and ethyl acetate (ERM) fractions showed significant free radical scavenging properties (IC50 value 12.18 and 11.7 μg/mL, respectively) compared to the standard ascorbic acid (6.36 μg/mL) in DPPH free radical scavenging assay. All three fractions exhibited notable analgesic effect in mice compared to standard drug diclofenac sodium in acetic acid induced writhing method. DRM and ERM fractions revealed significant glucose lowering effects compared to standard glibenclamide in streptozotocin induce diabetic mice model. Besides, all the fractions showed remarkable antibacterial effects (zone of inhibition 11.1-17.3 mm) against all selected Gram-positive but showed moderate activity against the Gram-negative bacteria. In-vitro cytotoxicity test of DRM and ERM fractions exhibited cytotoxic effect (IC50 value 88.94 µg/ml and 127.6 µg/ml, respectively) against HeLa cell. The in-vivo cell growth inhibition of the three fractions on EAC (Ehrlich ascites carcinoma) cell demonstrated that ERM fraction furnished maximum cell growth inhibition (54.61%) compared to 84.83% inhibition by bleomycin.,. From the above findings, it is evident that ethyl acetate fractions of R. mucronata can be exploited for future drug development and traditional medicinal applications.

2. The introduction needs to be further consolidated with scientific reports on the biological properties of this mangrove extracts. The authors should highlight how the current study is novel and adds to already reported literature. Consider publications such as Sadeer et al., 2019

Moreover, there are recent US patents on drug formulations derived from extracts of R. Mucronata. It is not clear why the authors chose only the leaf of the plant and did not consider other parts of the plant.

Author’s reply: Revised accordingly. Following texts have been added to introduction section in the manuscript.

Crude methanolic extract of Rhizophora mucronata has been reported to have anticancer properties in different cancer cell lines [9]. 80% methanolic extract of R. mucronata leaves possesses flavonoids, gallic acid, quercetin and coumarin that has antiradical, antihyperglycemic and reduced lipid peroxidation activities. [10]. Detailed assessment of the traditional uses such as anticancer, anti-inflammation, antimicrobial, antidiabetic etc along with 60 chemical components and their toxicity studies from R. mucronata has been reported earlier [11]. The antidiarrheal and anti-inflammatory activity guided isolation of bioactive components has led to isolate six compounds from R. mucronata bark extracts. [12] Tannins such as catechin, epigallocatechins were reported from the leaf extract of R. mucronata that relaxed or reduced the contractions of ileum in diarrheal rats. [13]

Please see also authors reply of query no 11

The leaf of the plant was chosen because of the following reason which has been included in the introduction section:

Leaf extracts of R. mucronata were chosen because very little investigation in the pharmacological activity of leaf extracts have been reported. Moreover, studies indicated that the high concentration fruit flour of R. mucronata contains trace amounts of hydrogen cyanide (HCN) leading to create toxicity [15]. Alternatively, fresh leave extract of R. mucronata is considered safe and possessed no hazardous effect at standard therapeutic concentration [16]

3. Regarding the in vivo anti-cancer study, can the authors provide a reference to support the protocol that was used.

Author’s reply: Thank you for your suggestion. The following write up has been mentioned in the manuscript along with referenced protocol:

In-vivo cell growth inhibition was carried out following the established protocol as described earlier [27]. Five groups of Swiss albino mice (n=5) were used for this study. (section 2.11)

4. The results should be discussed and compared with already reported data.

Author’s reply: Thank you for the suggestion. The results are compared with other data and the manuscript was revised accordingly. (Section 4; Discussion).

In case of antioxidant and antimicrobial properties, the results were already compared with the reported data (Section 4; discussion). In addition, the following sentences were added to compare the results of the antidiabetic, analgesic and anticancer effects:

In an earlier investigation, ethanolic extracts of R. mucronata showed 44% reduction in the writhing with a dose of 200 mg/kg body weight in mice which is in accordance with our findings [35]. The current extracts from Sundarbans showed better writhing inhibition.

Our findings reestablish the previous findings where ethanolic fractions of R. mucronata significantly inhibited α-amylase and α-glucosidase enzymes activity as well as controlled glycemic index and reduced diabetic complications [37].

These results reinforce the findings of a previous in vitro anticancer study of R. mucronate crude methanol extracts where the extracts were found to be effective against colon, lung, prostrate and breast cancers [9].

5. Consider depicting the results in the tables in the form of figures instead.

Author’s reply: Thank you. Some tables and figures have been changed ( previous Table 2 was transformed to Fig 2).

6. The figures are of poor quality.

Author’s reply: The quality and resolution of figures are improved accordingly.

7. Please conduct statistical analysis on figures 1 and 3.

Author’s reply: Statistical analysis was accomplished and mentioned in the text.

8. Provide the 1H-NMR spectra of the three isolated compounds and assign the peaks to the structure

Author’s reply: Thank you for the suggestion. We have provided NMR spectra in the supporting files and assigned the peaks as well. The complete spectral data is present in the manuscript.

9. Consolidate the discussion by linking the anti-cancer, anti-diabetic, anti-oxidant properties etc to the functional groups in the isolated compounds

Author’s reply: We have tried our best to incorporate few points on it. For example, in case of antioxidant and antimicrobial activity we have already included the following sentence:

Therefore, the highest activity of the comparatively polar ERM fraction indicates that polar compounds might be responsible for antioxidant effect. Plants are rich sources of flavonoids and phenolic compounds which have the ability to scavenge free radicals due to their ability to donate hydrogen atom [34]. Phytochemical investigation of ERM fraction provided phenolic compound N-trans-para-caffeoyl-tyramine (1) and flavonoids rutin (3) which might have contributed for the strong antioxidant property of ERM fraction.

The potent antimicrobial activity of the crude extracts may be due to the presence of phenols, flavonoids, alkaloids, steroids [43].

10. Define the abbreviations used in the abstract

Author’s reply: Abbreviations have been introduced accordingly.

11. They did not clearly point out the novelty of the current investigation.

Author’s reply: Novelty and Rational of the current investigation is described in Introduction and Conclusion section. The following sentences were added in introduction section which describe some novel approach for phytochemical and biological activity:

Although enormous investigation on R. Mocronata whole plant extracts has been accomplished previously, only few reports are found on different parts of the plant. Moreover, biological investigation on differential extracts or fractions depending on polarity of the solvents are missing. Moreover, comprehensive pharmacological screening including in vitro and in vivo anticancer studies along with detailed understanding and characterization of bioactive components are very rare. Leaf extracts of R. mucronata were chosen because very little investigation in the pharmacological activity of leaf extracts have been reported. (Section 1: Introduction)

The current study has been designed to investigate various pharmacological activities such as antioxidants, analgesic and anti-inflammatory, antidiabetic, antimicrobial and cytotoxic potentials with different fractions (HRM, DRM and ERM) of R. mucronata. Moreover, in this research we focused on isolation and characterization of active constituents from the active fraction of R. mucronata

Here three compounds N-trans-para-caffeoyl-tyramine (1) and β-sitosterol (2) and rutin (3) from ERM fraction of R. mucronata and N-trans-para-caffeoyl-tyramine (1) and rutin (3) were isolated from this plant for the first time. The fractions also gave significant in-vitro cytotoxic effect on HeLa cell and in-vivo anticancer effect against EAC cell which were per

---

## [Editor Report · Decision Letter 1]

26 Dec 2025

Pharmacological evaluation of mangrove plant Rhizophora mucronata (Lam.) grown in the coastal area of Sundarbans

PONE-D-25-31376R1

Dear Dr. Rahman,

We’re pleased to inform you that your manuscript has been judged scientifically suitable for publication and will be formally accepted for publication once it meets all outstanding technical requirements.

Kind regards,

Nadeem Nazurally, Ph.D

Academic Editor

PLOS One
---

## [Editor Report · Acceptance letter]

1 Jan 2026

PONE-D-25-31376R1

PLOS One

Dear Dr. Rahman,

I'm pleased to inform you that your manuscript has been deemed suitable for publication in PLOS One. Congratulations! Your manuscript is now being handed over to our production team.

Kind regards,

on behalf of

Dr. Nadeem Nazurally

Academic Editor

PLOS One